# Genome-Wide Identification of WD40 Proteins in *Cucurbita maxima* Reveals Its Potential Functions in Fruit Development

**DOI:** 10.3390/genes14010220

**Published:** 2023-01-14

**Authors:** Chen Chen, Yating Yang, Liu Pan, Wenhao Xia, Lanruoyan Xu, Bing Hua, Zhiping Zhang, Minmin Miao

**Affiliations:** 1College of Horticulture and Plant Protection, Yangzhou University, Yangzhou 225009, China; 2Joint International Research Laboratory of Agriculture and Agri-Product Safety of Ministry of Education of China, Key Laboratory of Plant Functional Genomics of the Ministry of Education/Jiangsu Key Laboratory of Crop Genomics and Molecular Breeding, College of Horticulture and Plant Protection, Yangzhou University, Yangzhou 225009, China

**Keywords:** *Cucurbita maxima*, WD40, fruit

## Abstract

WD40 proteins, a super gene family in eukaryotes, are involved in multiple biological processes. Members of this family have been identified in several plants and shown to play key roles in various development processes, including acting as scaffolding molecules with other proteins. However, WD40 proteins have not yet been systematically analyzed and identified in *Cucurbita maxima*. In this study, 231 WD40 proteins (CmWD40s) were identified in *C. maxima* and classified into five clusters. Eleven subfamilies were identified based on different conserved motifs and gene structures. The CmWD40 genes were distributed in 20 chromosomes; 5 and 33 pairs of CmWD40s were distinguished as tandem and segmental duplications, respectively. Overall, 58 pairs of orthologous WD40 genes in *C. maxima* and *Arabidopsis thaliana*, and 56 pairs of orthologous WD40 genes in *C. maxima* and *Cucumis sativus* were matched. Numerous CmWD40s had diverse expression patterns in fruits, leaf, stem, and root. Several genes were involved in responses to NaCl. The expression pattern of CmWD40s suggested their key role in fruit development and abiotic stress response. Finally, we identified 14 genes which might be involved in fruit development. Our results provide valuable basis for further functional verification of CmWD40s in *C. maxima*.

## 1. Introduction

WD40 proteins are regulators containing a highly conserved motif, also defined as WD repeat (WDR) or WD40 repeat domain. WD40s are rarely present in prokaryotes but are abundant in eukaryotes comprising 1–2% of proteins in the latter [1,2]. The N-terminus of the WD40 motif starts with a glycine-histidine (GH) pair, and the C-terminus ends with a tryptophan-aspartic acid (WD) pair [1,2]. Typically, a single repeat WD, consisting of 44–60 amino acids, contains a four-stranded anti-parallel β-sheet [3,4]. A bladed propeller scaffold contains five to eight such repeats and is the location where protein to protein interaction takes place [5,6,7,8,9,10,11]. The 7-fold β-propeller is predicted to be the most stable β-sheet based on geometry modeling and resolved WD40 structures are always dominated by this type of β-propeller [7,12,13]. Apart from the WD domain, the WD40s need additional domains to recruit other proteins and form protein complexes [7,8,9]. Therefore, the characteristics of these proteins are helpful for their interaction with other cellular components.

WD40 family members have been systematically identified in eukaryotes including several plant species [14,15,16]. Studies have reported 743 TaWD40s in wheat genome (*Triticum aestivum* L.) [14], 220 in *Prunus persica* L. [17], 579 in cotton (*Gossypium hirsutum*) [18], 187 in *Rosaceae* [19], 191 in cucumber (*C. sativus*) [20], 237 in thale cress (*A. thaliana*) [8], 200 in rice (*Oryza sativa*) [21], and 225 in foxtail millet (*Setaria italica*) [22]. To date, WD40 family members have not been systematically identified in *C. maxima*.

Regulatory homomeric or heteromeric complexes consist of protein–protein interaction, mainly regulated by scaffolding [4]. A common molecular mechanism of the action of WD40 proteins is that WD40 proteins participate in a series of cellular, physiological, and developmental processes as the scaffold of protein–protein interactions [23,24]. For example, as a component of the cytoplasm or nucleoplasm, WD40 proteins interact with membrane proteins and link to the cytoskeleton or membrane [17]. In plants, WD40 proteins bind with diverse proteins in extensive ways to form complex structures and are involved in various biological regulatory processes, such as cell movement and cycle, light vision and signaling, flowering transition, anthocyanin biosynthesis, meristematic recognition, and floral development [23,24,25,26,27,28,29,30,31,32,33,34,35,36,37]. TTG1, a WD40 protein in *A. thaliana*, *Antirrhinum majus*, and *Petunia hybrida*, acts as the scaffold for interaction with MYB and bHLH proteins and plays an important role in the regulation of trichome development, seed-coat pigmentation, epidermal cell, root-hair differentiation, etc. [38,39]. Among these functions of the WD40 protein in plants, WD40s regulate organ development via cell division and expansion. In *A. thaliana*, numerous WD40 proteins, such as WDR53-CesA8, GTP binding protein β1, Fasciata2 (FAS2), Leunig (LUG), Fertilization-Independent Endosperm (FIE), and Cyclophilin71 are involved in cellulose biosynthesis, cell division, meristem maintenance, floral development, seed development, and flowering, respectively [40,41,42,43,44]. In rice, OsLIS-L1, a five WD40-repeat protein participates in the regulation of internode elongation [45]. The germinating modulator of rice pollen (GORI), which encodes a WD40 protein, regulates pollen tube germination and elongation in rice [46], while OsKRN2 negatively regulates grain number in maize and rice [47]. Although a growing number of studies show that WD40 proteins are involved in the regulation of plant organ development, there are few studies indicating that WD40 proteins regulate the fruit development in *C. maxima*.

Atlantic Giant (AG) produces the world’s largest *C. maxima* fruit. Our previous work showed that CmaCh14G019450, encoding a WD40 repeat-like superfamily protein, was located in the QTL associated with fruit development of *C. maxima* [48]. Although this is significant for revealing the mechanism of fruit development in *C. maxima*, there are few studies exploring the role of WD40 proteins in fruit development of *C. maxima*. Here, we identified CmWD40 proteins members, including their exact number, ID, distribution in chromosome, phylogenetic analysis, gene structure, and duplication, in the *C. maxima* genome. We analyzed the syntenic relationships of WD40 proteins between *C. maxima*, *C. sativus*, and *A. thaliana.* The expression patterns of WD40 genes in different tissues were analyzed using transcriptome and qRT-PCR. Our systematic analysis suggested that proteins have a key role in fruit development and lays a foundation for further functional characterization of WD40 genes in fruit development.

## 2. Materials and Methods

### 2.1. Plant Materials and Treatment

The seeds of two varieties of *C. maxima*, Atlantic Giant (AG, the world’s largest *C. maxima*) and Hubbard (a cultivar with smaller fruit), were obtained from the Sustainable Seed Company (Covelo, CA, USA). Seeds were soaked in water at 60 °C for 30 min and then germinated at 28 °C. The germinated seeds were placed in soil in plastic pots. Seedlings were planted at the three-leaf stage in a plastic greenhouse at the Yangzhou University (119°26′ E, 32°24′ N); the plants were spaced 5 m apart and planted in rows spaced 2 m apart.

### 2.2. Acquisition and Identification of WD40 Proteins

Whole sequences of WD40 in *C. maxima* and *C. sativus* were downloaded from the CuGenDB database (http://cucurbitgenomics.org/ accessed on 8 August 2022), and genes containing WD40 were extracted using simple HMM search in TBtools [49]. Genes were extracted from *A. thaliana* and atypical or abandoned genes were removed [20]. Proteins containing more than three WD40 domains were kept, based on the predictions of the WDRR database (http://systbio.cau.edu.cn/wdrr/ accessed on 10 August 2022). Finally, protein sequences of *A. thaliana* were used as a query sequence for BLAST. SMART (https://smart.embl-heidelberg.de/ accessed on 10 August 2022) and WDRR databases were used to check whether the extracted sequences contained WD40 motifs.

### 2.3. Characterization of WD40 Protein in C. maxima

CuGenDB was used to analyze amino acid number and chromosome location. Compute pI/Mw tool for Expasy (https://web.expasy.org/compute_pi/ accessed on 11 August 2022) was used to analyze the protein molecular weight and pI. Instability index and hydropathicity were analyzed using the ProtParam tool in Expasy (https://web.expasy.org/protparam/ accessed on 11 August 2022). Transmembrane domain data were obtained from TMHMM 2.0 (https://services.healthtech.dtu.dk/service.php?TMHMM-2.0 accessed on 12 August 2022). The SignalP 3.0 tool (https://services.healthtech.dtu.dk/service.php?SignalP-3.0 accessed on 13 August 2022) was used to analyze signal peptide. Subcellular localization data were predicted using CELLO v. 2.5 (http://cello.life.nctu.edu.tw/ accessed on 14 August 2022).

### 2.4. Chromosome Distribution and Replication of the WD40 Gene Family

The genome sequence information of WD40s was obtained from CuGenDB; MapChart 2.32 was used to draw WD40 family genes on chromosomes. Ks values, Ka values, and Ka/Ks ratios of the WD40 gene family were estimated to assess evolutionary constraints. The Ka/Ks ratio was obtained using a simple calculator function in the TBtools software, based on the CDS and protein sequence of *C. maxima* and *C. sativus* obtained from CuGenDB and *A. thaliana* from the Arabidopsis genome database.

### 2.5. Collinear Analysis

The One Step MCScanX plugin in TBtools was used for collinearity analysis and the Advanced Circos program was used to visualize the collinearity relationship of WD40 family genes in *C. maxima*. The collinear relationship among *A. thaliana*, *C. maxima*, and *C. sativus* was visualized using the Multiple Synthetic Plot program [50].

### 2.6. Phylogenetic Analysis

The WD40 protein sequences of *A. thaliana*, *C. maxima*, and *C. sativus* WD40 were aligned using the MUSCLE algorithm of MEGA11 software. A phylogentic tree was constructed with the neighbor joining method with 1000 bootstrapped replicates. The evolutionary distances were computed using a Poisson correction model and pairwise deletion. The final phylogenetic trees were visualized using the online tool iTol.

### 2.7. Gene Structure, Conserved Motifs, and Promoter Region Analysis

The WD40 gene structure information was extracted from CuGenDB and conserved domains were predicted by the NCBI Batch Web CD-Search tool (https://www.ncbi.nlm.nih.gov/Structure/bwrpsb/bwrpsb.cgi? accessed on 16 August 2022). MEME v.5.4.1 (https://meme-suite.org/meme/tools/m-eme accessed on 17 August 2022) was used to identify conserved motifs, with parameters such as 10 motifs and other default values. Sequences 2000 bp upstream of the *C. maxima* coding region were obtained from CuGenDB. These sequences were submitted to the online tool PlantCARE (http://bioinformatics.psb.ugent.be/webtools/plantcare/html/ accessed on 19 August 2022) to predict the promoter cis-acting element. The TBtools software BioSequence Structure Illustrator was used for the visualization.

### 2.8. Expression Analysis

Gene expression date in diverse organs, including fruit, leaf, stem, root (PRJNA385310), stem phloem, and vascular tissues (SRP012853), as well as *Cucurbita moschata* cv. N12 leaf mesophyll and vein under control or NaCl treatment (PRJNA464060), were retrieved from CuGenDB (http://cucurbitgenomics.org/ accessed on 20 August 2022). A heatmap was drawn using the HeatMap Illustrator program in TBtools [50].

### 2.9. RNA Extraction and Quantitative Real-Time-PCR

Total RNA was isolated from fruit, leaf, stem, and root, using the Trizol RNA Kit (TaKaRa, Dalian, China) following the manufacturer’s protocol. RNA integrity was checked using Agarose gel electrophoresis. The proposed RNA was reverse recorded with EasyScript^®^ One-Step gDNA Removal and cDNA Synthesis SuperMix Kit (TRANS, Beijing, China). qRT-PCR assays were performed, using the iTaq Universal SYBR^®^ Green Supermix Kit (Bio-Rad Laboratories, Hercules, CA, USA) on the CFX-96 (Bio-Rad Laboratories, Hercules, CA, USA). The primers for this study, derived from the NCBI Primer-BLAST online tool (https://www.ncbi.nlm.nih.gov/tools/primer-blast/index.cgi?LINK_LOC=BlastHome accessed on 21 August 2022) were tested using CFX-96 (Bio-Rad Laboratories, Hercules, CA, USA) and listed in Appendix A.

## 3. Results

### 3.1. Identification of WD40 Genes in C. maxima

Totally, 237 *A. thaliana* WD40 genes (AtWD40s) were obtained from TAIR (http://www.arabidopsis.org/ accessed on 22 August 2022). From these, 18 genes were removed for their atypical structure and lack of WD40 domain based on the simple HMM Search and WDRR (http://systbio.cau.edu.cn/wdrr/ accessed on 23 August 2022) and 203 typical WD40 genes were used for further analysis. Further, the WD40 proteins were initially obtained using the 203 AtWD40 proteins as the query for BLAST. Finally, 260 and 187 putative WD40 proteins were identified in *C. maxima* and *C. sativus*. We removed the WD40 proteins without WD40 motifs using online tools of SMART and WDRR. Finally, a total of 203, 172, and 231 genes were identified in *A. thaliana*, *C. sativus*, and *C. maxima*, respectively (Appendix A).

The lengths of the coding sequence (CDS) of the 231 CmWD40 genes showed wide variation, ranging from 681 to 5256 bp with an average length of approximately 2287.5 bp. The length of *C. maxima* WD40 proteins (CmWD40) ranged from 226 to 1751 amino acid residues (aa), with an average length of 988.5 aa approximately. Silico analysis revealed that the relative molecular mass of the WD40 family protein was in the range of 24996.96 Da to 194453.58 Da, and the average molecular mass was 109725.27 Da. The isoelectric point (pI), instability index, and hydropathicity were also analyzed to evaluate the physicochemical properties of WD40 proteins. The theoretical pI of the CmWD40 proteins ranged from 4.44 to 9.73; 78 CmWD40 proteins had pI greater than 7.085. The instability index of CmWD40 proteins varied from 24.63 to 62.45 with an average instability index of 43.54. Among the CmWD40s, a total of 142 WD40 protein sequences are considered to be unstable, and 89 are considered to be stable (Appendix A). The grand average of hydropathicity (GRAVY) ranged from −0.872 to 0.161, and the average value was −0.6545 (Appendix A).

### 3.2. Protein Classification and Phylogenetic Analysis of the C. maxima WD40 Gene Family

To gain insights into the functional divergence and the evolutionary relationship of WD40 proteins in *A. thaliana*, *C. sativus*, and *C. maxima*, full length protein sequences were used to construct phylogenetic trees using the Mega11 software with the neighbor joining method followed by the MUSCLE algorithm. Based on the results of phylogenetic tree analysis, the WD40 proteins could be grouped into five clades. As shown in Figure 1, there were 87, 90, 154, 135, and 140 WD40 genes in the I–V clades, including 32, 39, 57, 51, and 52 *C. maxima* WD40 genes, respectively. Significantly, *CmaCh14G019450*, a WD40 gene identified and located in the QTL associated with fruit development, was located in the third clade (Figure 1).

To further explore the divergence of the CmWD40s, the 231 CmWD40 proteins were divided into 11 subfamilies based on their domain compositions (Appendix A). Among these, subfamily A, the CmWD40 proteins with only WD40 domain, contained the most members (138). Both subfamily H (WD40 domain and BEACH domain) and subfamily J (WD40 and breast carcinoma amplified sequence 3 domain) only contained one member (Appendix A). I clade contains 26 members of subfamily A, 1 member of subfamily H, member of subfamily I, and 4 members of subfamily K; II clade contains 20 members of subfamily A, 3 members of subfamily B, 9 members of subfamily D, 1 member of subfamily F, 6 members of subfamily K, III clade contains 28 members of subfamily A, 2 members of subfamily B, 3 members of subfamily C, 7 members of subfamily E, 1 member of subfamily F, 3 members of subfamily G, 1 member of subfamily I, and 12 members of subfamily K; IV clade contains 31 members of subfamily A, 8 members of subfamily B, 4 members of subfamily C, and 8 members of subfamily K; V clade contains 35 members of subfamily A, 2 members of subfamily D, 1 member of subfamily I, 1 member of subfamily J, and 13 members of subfamily K.

### 3.3. Chromosomal Distribution, Gene Replication Analysis of CmWD40s

To identify the genomic distribution and gene replication of CmWD40s, we used MapChart 2.32 software for further analysis. Overall, 231 WD40 genes were mapped onto chromosomes 1 to 20 (Chr1-Chr20) of the *C. maxima* genome in an uneven manner; CmWD40 genes ranged from 5 to 25 in these chromosomes. Chr11 and Chr16 contain the largest and smallest numbers of CmWD40 genes, respectively. The *CmaCh14G019450* was located in Chr14. The CmWD40 genes distribution distances varied from 3.04 MB in Chr10 to 19.66 MB in Chr4 (Figure 2).

Gene family expansion and evolution are always accompanied by frequent tandem and segmental duplications, leading to gene clusters and scattered family members, respectively. To determine the gene replication events in *C. maxima*, we conducted a collinear analysis using the One Step MCScanX plugin in TBtools. Based on the results of collinear analysis of CmWD40 genes, 33 gene pairs of duplicated segments were identified with five tandem duplications within the *C. maxima* genome. The gene duplication analysis showed that the segmental duplication events were more vital than tandem duplication in the expansion and evolution of the WD40 family in *C. maxima* (Figure 3).

To further estimate the evolutionary dates and reveal a functional selection pressure between duplicated gene pairs, the non-synonymous (Ka) and synonymous (Ks) parameters were used to calculate gene duplications. The (Ka)/ (Ks) values for all 33 duplicated CmWD40 gene pairs were less than 1, suggesting that the CmWD40 genes were mainly under purifying selection during the family evolution and expansion (Appendix A). Therefore, the duplication events of CmWD40s play a key role in the evolution and expansion of the WD40 family in *C. maxima*. The syntenic relationships of the WD40 family members of *C. maxima*, *C. sativus*, and *A. thaliana* were further explored to gain insight into the evolution of WD40s in these three species. The results of the multi-species collinearity analysis showed 58 pairs of paralogs between *A. thaliana* and *C. maxima* and 56 pairs of paralogs between *C. maxima* and *C. sativus* (Figure 4). Thus, in the WD40 gene family, the relationship between *C. maxima* and *A. thaliana* is slightly closer.

### 3.4. Gene Structure and Promoter Region Analysis

Previous studies show that gene structure diversity always causes expansion and evolution of the gene family. To further characterize the gene structure diversity of CmWD40s, we conducted a motif analysis to reveal the diversity of WD40 proteins and their functional divergence in *C. maxima*. The candidate gene *CmaCh14G019450* was grouped in Clade III; thus 57 *C. maxima* WD40 proteins of Clade III (CmWD40-III) were selected for further study. The MEME tool was used to characterize the motifs and determine the predicted structural characteristics of CmWD40-III (Figure 5). Nine motifs were identified and were unevenly distributed in the CmWD40-III proteins (Figure 5). Most members of the CmWD40-III proteins contained motif 1 and 4 and less than half of the CmWD40-III proteins contained motif 5, 6, 7, and 8. Interestingly, the phylogenetic tree analysis revealed that most CmWD40-III proteins containing motif 5 were grouped in one cluster, and the CmWD40-III proteins belonging to one cluster contained similar motifs. We also analyzed the conserved domains of CmWD40 proteins, and 10 conserved domains (WD40, Abhydrolase, Atrophin-1, DUF, LisH, NLE, RING, S_TKc, Ubox, and Utp12 domain) were identified. The results of conserved domain analysis show that the proteins belonging to each cluster contained similar conserved domains, such as similar number and location of WD40 domains. To better understand the structural diversity of CmWD40 genes, we analyzed the exons and introns in the genes. The numbers of exons and introns ranged from 1~30 and 0~29, respectively. These results suggest that the gene structures of CmWD40 were diverse, and members of one cluster might have similar functions for similar protein structures (Figure 5).

Cis-elements in the promoter region regulate gene expression by interacting with their corresponding transregulatory factors and contributing to plant development, phenotypic evolution, adversity adaptability, etc. To better explore the regulation of gene expression and the WD40 function, the 2000 bp genomic sequences upstream of the translation start site were defined as the promoter region and the cis-elements in the promoter regions of the CmWD40s were analyzed using the PlantCARE database (http://bioinformatics.psb.ugent.be/webtools/plantcare/html/ accessed on 23 August 2022). The analysis shows that a series of cis-elements involved in low-temperature responsiveness (CCGAAA), salicylic acid responsiveness (CCATCTTTTT), abscisic acid responsiveness (ACGTG), anaerobic induction (AAACCA), light responsiveness (TACGTG or CACGAC), MeJA-responsiveness (TGACG or CGTCA), meristem expression (GCCACT), and endosperm expression (TGAGTCA) were identified; most of these were related to light responsiveness. The results of the cis-elements analysis indicated that the CmWD40 genes were putatively regulated at the transcriptional-level by light (Figure 5).

### 3.5. The Gene Expression Pattern Analysis of CmWD40s

Previous studies show that several WD40 proteins are involved in cell division and expansion [45]. Therefore, WD40 genes that were preferentially expressed in fruit might be involved in fruit development. Besides those genes related to cell division and expansion, the balance of source–sink also contributes to fruit development. Thus, WD40 genes highly expressed in leaf and fruit might contribute to the balance of source–sink, which plays a key role in the fruit development. To identify if WD40 genes were highly expressed in leaf and fruit, we analyzed the expression of CmWD40s genes in different tissues using the RNA-seq date in diverse organs downloaded from CuGenDB. As shown in Figure 6, several genes were highly expressed in different organs; 17 WD40 genes were highly expressed in fruit (the ratio of expression in fruit to the expression in leaf was greater than 2.5); there were 28 WD40 genes highly expressed in leaf (the ratio of expression in leaf to the expression in fruit was greater than 3) (Appendix A). Furthermore, we analyzed the WD40 expression in mesophyll and vein tissues of leaves and found two WD40 genes highly expressed in mesophyll (the ratio of expression in mesophyll to vein was greater than 2) and 22 genes highly expressed in veins (the ratio of expression in vein tissue to mesophyll was greater than 2) (Appendix A). Additionally, 11 WD40s were highly expressed in leaf and vein tissue (Appendix A). Therefore, we selected five WD40 genes which were highly expressed in fruit (CmaCh09G006530, CmaCh09G005870, CmaCh18G007320, CmaCh04G017900, and CmaCh08G003130), five genes in vein tissues (CmaCh03G004280, CmaCh04G004520, CmaCh02G016250, CmaCh15G003800, and CmaCh04G026570), and two genes highly expressed in the mesophyll of leaf (CmaCh18G003250 and CmaCh18G007320) for further analysis. We previously found 23 WD40s expressed significantly differently in AG and Hubbar, with *CmaCh01G002640* and *CmaCh09G008800* being most different. Our previous work showed that *CmaCh14G019450* had a putative role in the regulation of *C. maxima* fruit development (Figure 7; Appendix A; Yamagishi et al., 2005; Pan et al., 2022). Combining the transcriptome and the function of homologous genes, we selected 14 CmWD40 genes and characterized the expression of those genes in fruit, leaves, stems, and flowers of AG and Hubbard using qRT-PCR assay. In generally, the results of qRT-PCR assay were consisted with transcriptome (Figure 6). In additionally, the qRT-PCR results showed six genes highly expressed in fruit (*CmaCh09G006530, CmaCh09G005870, CmaCh18G007320, CmaCh04G017900, CmaCh09G008800*, and *CmaCh01G002640*) that were significantly differently more expressed in the leaves of AG than Hubbard; six genes (*CmaCh09G006530, CmaCh09G005870, CmaCh18G007320, CmaCh18G003250, CmaCh04G026570*, and *CmaCh01G002640*) were significantly differently more expressed in the fruits of AG and Hubbard (Figure 6). Thus, those eight CmWD40 genes were differently expressed in AG than Hubbard, indicating that these genes potentially had roles in the regulation of fruit development.

## 4. Discussion

WD40 proteins play key roles in the regulation of plant development. The present study systematically identified and characterized the WD40 genes in *C. maxima*. A total of 231 CmWD40 proteins were identified and clustered into five subfamilies according to the phylogenetic analysis: I (32 genes), II (39 genes), III (57 genes), IV (51 genes), and V (52 genes). The results were supported by conserved domain and gene structural analysis. The CmWD40s were unevenly distributed in chromosomes 1 to 20 in the *C. maxima* genome, and the tandem and segmental duplication of CmWD40s was distinct, suggesting that gene duplication was important for the expansion of the *C. maxima* WD40 gene family. The results of transcriptome sequencing and qRT-PCR revealed that 14 genes had a potential role in the regulation of fruit development in *C. maxima*. Our previous works show that the *CmaCh14G019450* gene was involved in the regulation of fruit development and our current results are consistent with the analysis of this study.

Plant hormones play key roles in organ development, and auxin and gibberellin are key hormones in fruit development [50,51,52]. Cis-elements in the gene promoter are involved in gene regulation by interacting with special trans-regulatory factors [53]. Our analyses showed that a lot of cis-elements existed in the promoters of WD40s, and auxin- and gibberellin-responsive elements were identified in the WD40s promoters. These results suggest that WD40 genes might be regulated by auxin and gibberellin. There were eight auxin-responsive elements and 14 gibberellin-responsive elements in the promoter of 14 WD40 genes. Interestingly, there are two gibberellin-responsive elements in the promoter of the *CmaCh14G019450* gene, indicating that *CmaCh14G019450* might be involved in the development of gibberellin. The transcriptome analysis and qRT-PCR results showed that there were several WD40 genes highly expressed in mature leaves of AG than H (Figure 6 and Figure 7). There were 84 cis-elements involved in light responsiveness existed in the promoter of 14 candidate WD40 genes. Light signal always contributes to the metabolism of assimilates, which play a vital role in the regulation of source–sink [54]. Therefore, the WD40 genes might be regulated by light signals and regulate the metabolism of assimilates and the source–sink relationship.

In *A. thaliana*, *EFO1* (*AT5G52250*) and *EFO2* (*AT5G23730)* play overlapping roles in the development of hypocotyl and the plant organs [55]. The knockdown of *SWA1* (*AT2G47990*) significantly inhibits root growth and is essential for the progression of the mitotic division cycles during gametogenesis in *A. thaliana* [56]. The *alp2* (*AT4G29860*) mutant caused defects in both embryo and seedling development [57]. The mutants of *LRS1* (*AT3G05090*) show defects in lateral root development and affect the auxin accumulation during lateral root primordium development and meristem emergence [58]. Interestingly, these genes were grouped in the cluster III, indicating that the genes in this cluster might contribute to organ development in plants. In *C. maxima*, *CmaCh05G011780* was homologous to *ALP2* and *LRS1*, and *CmaCh17G011410* was to *EFO1*, *EFO2*, and *SWA1*. Apart from these two genes, *CmaCh11G002620, CmaCh09G008650, CmaCh01G012080, CmaCh18G005880, CmaCh20G002360* were also present in this cluster.

## 5. Conclusions

Taken together, 231 WD40 proteins (CmWD40s) were identified in *C. maxima* and 14 CmWD40s were presumably involved in fruit development through the combined expression analysis.

In this study, 231 WD40 proteins (CmWD40s) were identified in *C. maxima* and classified into five clusters. Eleven subfamilies were identified based on different conserved motifs and gene structures. The CmWD40 genes were distributed in 20 chromosomes; 5 and 33 pairs of CmWD40s were distinguished as tandem and segmental duplications, respectively. Totally, 58 pairs of orthologous WD40 genes in *C. maxima* and *A. thaliana*, and 56 pairs of orthologous WD40 genes in *C. maxima* and *C. sativus* were matched. Numerous CmWD40s had diverse expression patterns in fruits, leaf, stem, and root. Several genes were involved in responses to NaCl. The expression pattern of CmWD40s suggested their key role in fruit development and abiotic stress response. Finally, we identified 14 genes which might be involved in fruit development. Our results provide valuable basis for further functional verification of CmWD40s in *C. maxima*.

## Figures and Tables

**Figure 1 genes-14-00220-f001:**
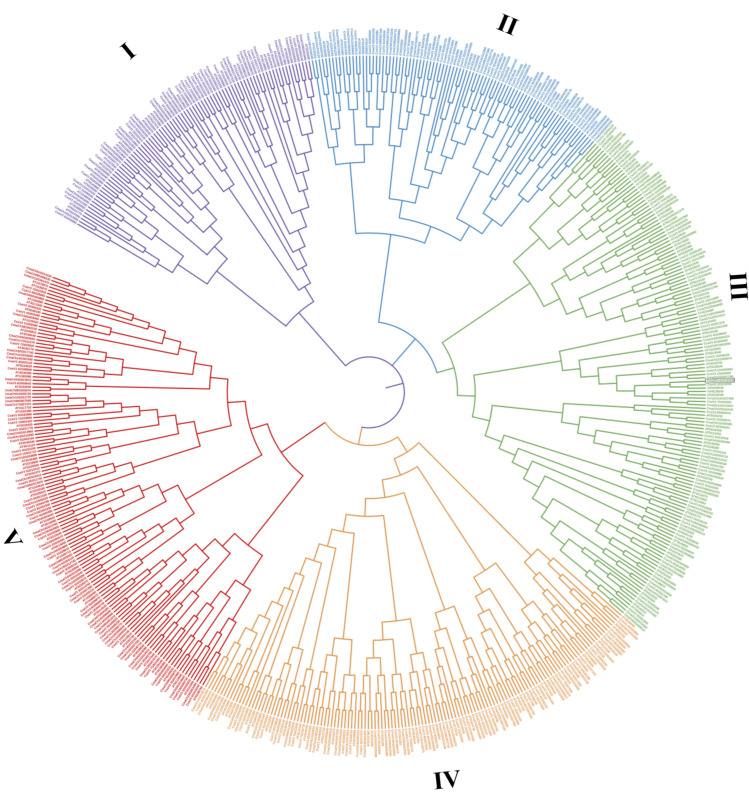
Phylogenetic relationships of the WD40 gene family proteins of *C. maxima*, *A. thaliana*, and *C. sativus*. The tree was constructed using MEGA11. The 231 *C. maxima* proteins, 203 *A. thaliana*, and 172 *C. sativus* proteins were divided into five clusters, labelled with different colors.

**Figure 2 genes-14-00220-f002:**
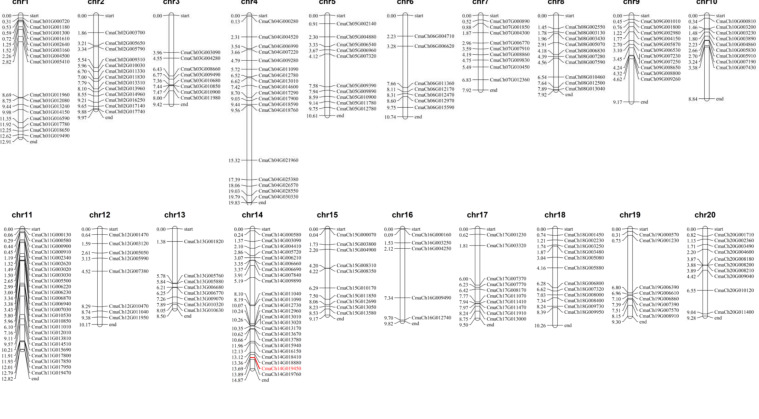
Uneven distribution of WD40 genes in *C. maxima* chromosomes. The bars represent chromosomes, and the black line on the olive bars indicate the location of WD40 genes on chromosomes. The number of chromosomes is represented at the top of each chromosome. Scale bar is in Megabase (Mb). The red part indicates the location of *CmaCh14G019450* in Chr14.

**Figure 3 genes-14-00220-f003:**
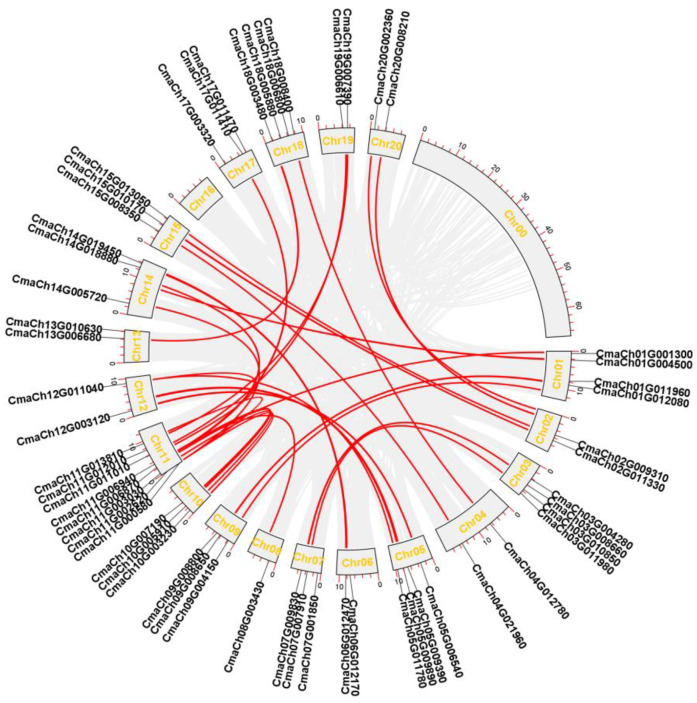
Distribution and segmental duplication of WD40 genes in *C. maxima.* The grey panel shows 21 chromosomes using a circle, red lines connect homologous genes; WD40 genes are marked outside of the circle.

**Figure 4 genes-14-00220-f004:**
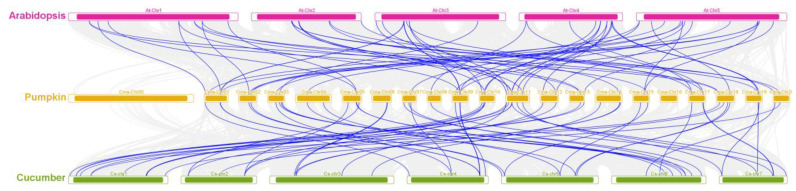
Synteny analysis of *C. maxima* WD40 genes between *A. thaliana* and *C. sativus*. The grey lines in the background indicate the collinear blocks between *C. maxima* and other plant genomes, while the blue lines highlight the syntenic WD40 gene pairs. The chromosome number is indicated at the top of every chromosome.

**Figure 5 genes-14-00220-f005:**
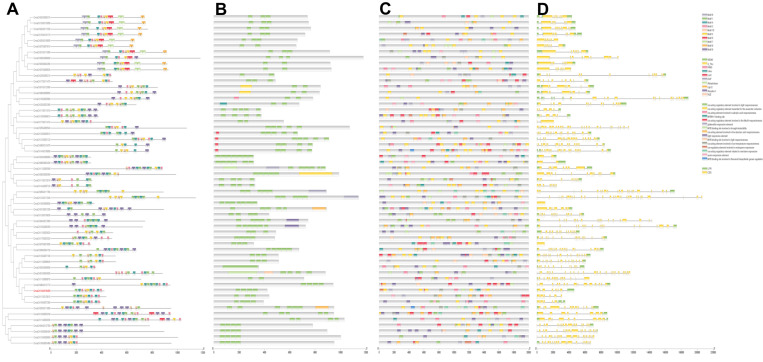
Phylogenetic analysis, conserved motifs, domains, promoter region analysis, and gene structure of WD40s. (**A**) Phylogenetic analysis of 57 Clade III *C. maxima* WD40s and conserved motifs of WD40s. Boxes of different colors represent 10 putative motifs. (**B**) Diversified protein domains in WD40s. The WD40 repeats are in green, while the other domains are marked in different colors. (**C**) Cis-elements in the promoter region of WD40s. (**D**) Exon/intron organization of WD40s. UTRs are green rectangles, exons are indicated in yellow rectangles, and the black line connecting two exons represent introns. The image was made with TBtools.

**Figure 6 genes-14-00220-f006:**
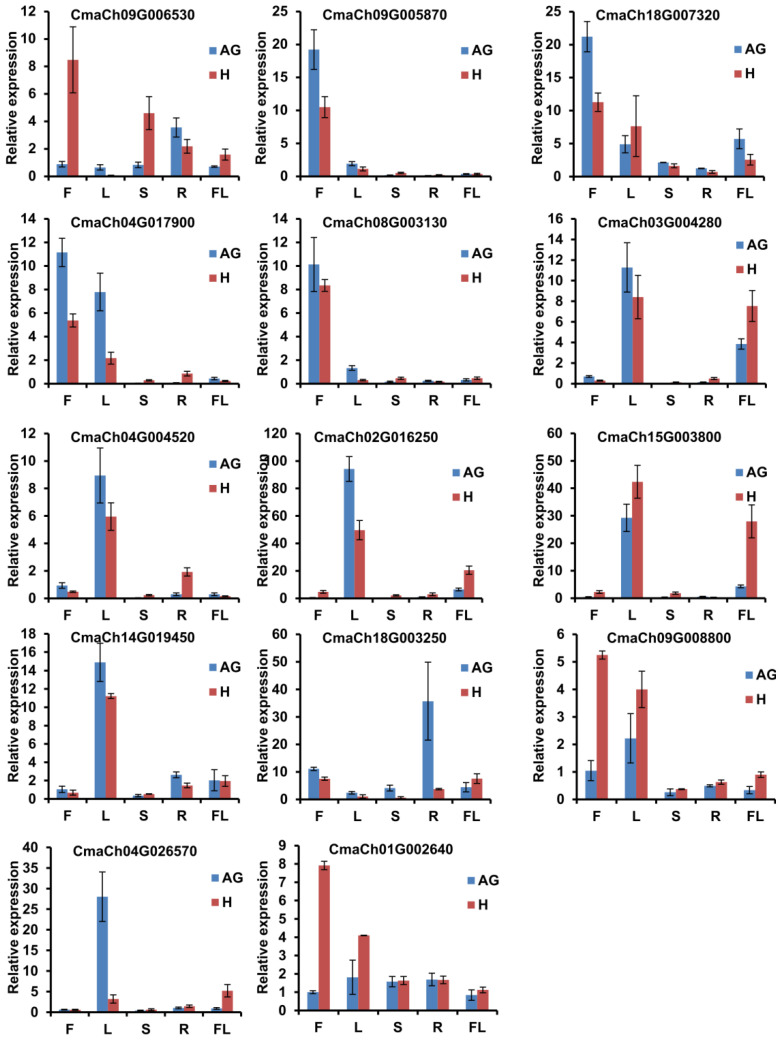
Expression analysis of CmWD40 genes in different tissues of Atlantic Giant (AG) and Hubbard (H). F (Fruit at 30 days after flowering), L (Mature leaves), S (Stems near the fruit), and FL (Flower buds) were used to characterize. The gene expression levels are normalized to the internal control of CmEF. The error bars represent the standard deviation of three biological replicates.

**Figure 7 genes-14-00220-f007:**
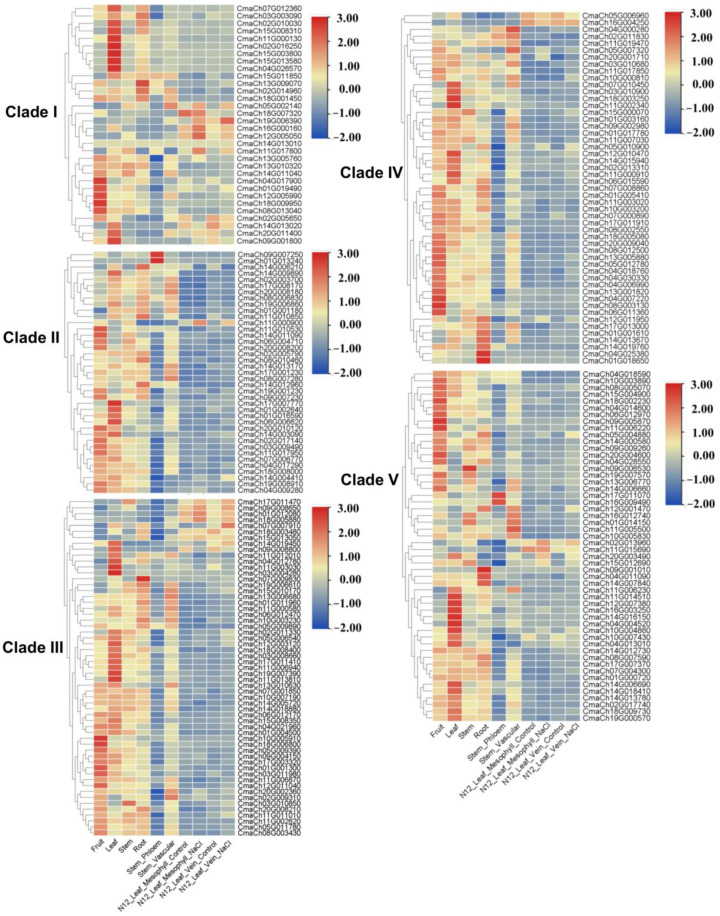
Heatmap of WD40 genes’ specific expression in leaf, root, fruit, stem, N12 leaf mesophyll, and vein after NaCl treatment. The data of expression in diverse organs downloaded from CuGenDB website can be seen in Appendix A.

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
