# Peer review of "Genome-Wide Identification of WD40 Proteins in Cucurbita maxima Reveals Its Potential Functions in Fruit Development"

_genes, 2023, doi:10.3390/genes14010220_

Round 1

Reviewer 1 Report

I have reviewd the mansucript. It is generally well written. I would suggest the authors to provide figures in a better resolution, since it is difficult to see what they represent. 

I also suggest adding the Conclusion section, with important findings and the novelty of this study. 

Author Response

Thanks for the comments, and we revised the manuscripts accordingly.

Reviewer 2 Report

Dear Sir/Madam,

In this MS, authors identified and characterized the WD40 proteins in C. maxima. They also analyzed the expression levels of the genes in various tissues including vegetative tissues and fruit. Generally, the analyses and experiments are good in the study. However, some issues attached should be clarified before it is accepted. 

Author Response

(The authors gave the same response as above.)

Reviewer 3 Report

The authors perform a Genome‑wide identification of WD40 proteins in Cucurbita 2m axima reveals and it's potential functions in fruit development.

The results indicate that identi The fied 14 genes which might be involved in fruit development and results provide valuable basis for functional verification of CmWD40s in C. maxima.

my point of view the work is well planned and resolved.

The results and the discussion are clear enough.

Author Response

(The authors gave the same response as above.)
